# Can Public Spaces Effectively Be Used as Cleaner Indoor Air Shelters during Extreme Smoke Events?

**DOI:** 10.3390/ijerph18084085

**Published:** 2021-04-13

**Authors:** Amanda J. Wheeler, Ryan W. Allen, Kerryn Lawrence, Christopher T. Roulston, Jennifer Powell, Grant J. Williamson, Penelope J. Jones, Fabienne Reisen, Geoffrey G. Morgan, Fay H. Johnston

**Affiliations:** 1Mary MacKillop Institute for Health Research, Australian Catholic University, Melbourne 3000, Australia; 2Menzies Institute for Medical Research, University of Tasmania, Hobart 7000, Australia; Penelope.Jones@utas.edu.au (P.J.J.); Fay.Johnston@utas.edu.au (F.H.J.); 3Faculty of Health Sciences, Simon Fraser University, Burnaby, BC V5A 1S6, Canada; allenr@sfu.ca; 4Mid North Coast Local Health District, Port Macquarie 2444, Australia; kerryn.lawrence@health.nsw.gov.au; 5Climate Science Centre, CSIRO Oceans and Atmosphere, Aspendale 3195, Australia; chris.roulston@csiro.au (C.T.R.); Jennifer.Powell@csiro.au (J.P.); Fabienne.Reisen@csiro.au (F.R.); 6School of Natural Sciences, University of Tasmania, NSW Bushfire Risk Management Research Hub, Hobart 7000, Australia; Grant.Williamson@utas.edu.au; 7Sydney School of Public Health, and University Centre for Rural Health, University of Sydney, Sydney 2006, Australia; geoffrey.morgan@sydney.edu.au

**Keywords:** smoke, indoor air quality, interventions, cleaner indoor air shelter, HEPA, PM_2.5_, disaster

## Abstract

During extreme air pollution events, such as bushfires, public health agencies often recommend that vulnerable individuals visit a nearby public building with central air conditioning to reduce their exposure to smoke. However, there is limited evidence that these “cleaner indoor air shelters” reduce exposure or health risks. We quantified the impact of a “cleaner indoor air shelter” in a public library in Port Macquarie, NSW, Australia when concentrations of fine particulate matter (PM_2.5_) were elevated during a local peat fire and nearby bushfires. Specifically, we evaluated the air quality improvements with central air conditioning only and with the use of portable high efficiency particulate air (HEPA) filter air cleaners. We measured PM_2.5_ from August 2019 until February 2020 by deploying pairs of low-cost PM_2.5_ sensors (i) inside the main library, (ii) in a smaller media room inside the library, (iii) outside the library, and (iv) co-located with regulatory monitors located in the town. We operated two HEPA cleaners in the media room from August until October 2019. We quantified the infiltration efficiency of outdoor PM_2.5_ concentrations, defined as the fraction of the outdoor PM_2.5_ concentration that penetrates indoors and remains suspended, as well as the additional effect of HEPA cleaners on PM_2.5_ concentrations. The infiltration efficiency of outdoor PM_2.5_ into the air-conditioned main library was 30%, meaning that compared to the PM_2.5_ concentration outdoors, the concentrations of outdoor-generated PM_2.5_ indoors were reduced by 70%. In the media room, when the HEPA cleaners were operating, PM_2.5_ concentrations were reduced further with a PM_2.5_ infiltration efficiency of 17%. A carefully selected air-conditioned public building could be used as a cleaner indoor air shelter during episodes of elevated smoke emissions. Further improvements in indoor air quality within the building can be achieved by operating appropriately sized HEPA cleaners.

## 1. Introduction

The health impacts of exposure to fire-related smoke pollution are well established. These include increases in asthma incidence, use of rescue medication, hospitalisations and emergency department visits. Fire-related smoke pollution also exacerbates and causes cardiovascular disease, and can result in increased mortality [1,2,3]. Around one-third of the Australian population is at elevated risk of developing smoke-associated illnesses during extreme smoke events [2]. These include individuals with respiratory and cardiovascular diseases, the very young, and older adults [2]. Particulate matter (PM) is the component of smoke most strongly and consistently linked with adverse health effects [4,5].

Standard recommendations to reduce exposure to smoke include the following: avoiding strenuous outdoor exercise, staying indoors with the doors and windows closed, using the air conditioning, and considering visiting an air conditioned public facility [6,7]. However, this advice is based on limited evidence of effectiveness for reducing personal exposure to air pollution [8].

A substantial fraction of outdoor air pollution makes its way indoors. In fact, most exposure to outdoor-generated fine particulate matter (PM_2.5_) occurs indoors and most of the health impacts from outdoor-generated PM_2.5_ are attributable to exposure that occurs while indoors [9,10]. Exposures that occur indoors account for 61% and 81% of the deaths attributed to outdoor-generated PM_2.5_ in the USA and China, respectively [11,12]. 

The infiltration efficiency of outdoor air pollution (*F_inf_*) is defined as the fraction of the outdoor PM_2.5_ concentration that penetrates indoors and remains suspended under steady-state conditions. *F_inf_* depends on the particle penetration efficiency, the air exchange rate, and the deposition rate. Methods to reduce building infiltration efficiency include closing windows and doors and filtering the air [9,13]. 

One of the most promising indoor defenses against outdoor smoke is the use of portable high efficiency particulate air (HEPA) filters, either in the building’s heating, ventilation, and air conditioning (HVAC) system or in portable air cleaner units. HEPA filters capture ≥ 99.97% of 0.3 µm particles. In previous studies, HEPA cleaners reduced residential PM_2.5_ concentrations by 52–67% when used for between 2–3 days and up to 2 weeks in US and Canadian communities impacted by woodsmoke [13,14,15,16]. Some public health agencies in North America recommend use of portable air cleaners to mitigate the impacts of forest fire smoke in residential settings [4]. 

The efficacy of portable air cleaners in larger public spaces has yet to be fully evaluated. Public spaces are increasingly being considered for providing access to “cleaner indoor air spaces” during smoke-related episodes [17,18]. However, more information is needed about the protection afforded by air-conditioned spaces and the potential for portable air cleaners to provide additional protection.

The Lindfield Park Road bushfire in Port Macquarie, New South Wales, Australia was one of the fires that heralded the beginning of Australia’s most extreme and prolonged bushfire and air pollution crisis on record. The fire began in mid-winter (July 2019) and moved into peat and continued to burn until February 2020. This fire, combined with many other bushfires in the area, caused several extreme and prolonged outdoor smoke episodes in the region. Peat fires are rarely extinguished rapidly, and the associated air pollution is likely to be prolonged. The emergency response agencies considered whether specific public buildings should be nominated as places to offer some respite from smoke pollution for the general public. Port Macquarie Hastings Council agreed to allow air quality monitoring stations to be set up both inside and outside the local library to evaluate the effectiveness of the building as a cleaner indoor air space. 

The aims of the research were as follows:(1)To evaluate the potential for a public building to serve as a cleaner indoor air shelter during smoke events, and(2)To assess the efficacy of installing HEPA cleaners within a smaller room inside the library area.

## 2. Materials and Methods

The selection of the Port Macquarie library for assessment as a potential “cleaner indoor air shelter” was made after considering a number of public buildings for a range of factors including the following: access, presence of air-conditioning, access to wifi, sufficient seating for the public, and access to public toilets among other criteria (see Appendix A for the checklist tool which was developed specifically for this occasion).

We measured PM_2.5_ concentrations at the Port Macquarie library from August 23rd, 2019 until November 29th, 2019 using Smoke Observation Gadgets, (SMOG) developed by the Commonwealth Scientific and Industrial Research Organisation (CSIRO). The low-cost sensors include a Plantower 3003 PM sensor (model PMS3003, Plantower, Beijing, China). The SMOG devices logged PM_2.5_ concentrations at 1-minute resolution. We deployed pairs of SMOG instruments (i) inside the main library, (ii) in the 22 m^2^ (3.7 m by 6.0 m) media room within the library, (iii) outside in the courtyard, and (iv) collocated with the NSW Department of Planning Industry and Environment’s Office of Environment and Heritage (DPIE) mobile monitoring station in the town. Explanatory signage was erected next to the indoor SMOGs. 

SMOG data were corrected using the calculations generated from collocated units with the DPIE managed 1405-DF TEOM located in town. This unit received regular maintenance during the campaign and provides a self-referencing NIST-traceable true mass measurement through internal collected filters. A 3rd order polynomial fit was applied to the SMOG data and the limit of detection for the SMOG units was 5.5 µg/m^3^. Direct comparisons of the two SMOG units at each of the tested locations demonstrated a strong association across the range of PM_2.5_ concentrations, see Appendix A.

We placed two portable HEPA cleaners (Cli-Mate AP20) (Aquaport Corporation Pty Ltd., Torrensville, SA, Australia) in the library’s media room and operated them on the medium fan setting from 24th August until 16th October 2019. These had a Grade H12 HEPA filter and the capacity to clean 20 m^2^ rooms with a smoke clean air delivery rate (CADR) of 133 m^3^/h. The selection of the HEPA cleaners and the space to be cleaned, needs to account for their operational capacity. This, therefore, restricted their use to a smaller room within the main library. The units were returned to have the filters replaced after this date so were not available for the remainder of the data collection period.

Because our primary interest was in quantifying relationships between indoor and outdoor PM_2.5_ concentrations during high smoke concentration events, we included in our analysis only periods with 24-h mean outdoor PM_2.5_ concentrations ≥ 20 µg/m^3^ and hours with valid measurements indoors and outdoors. This left 749 h of data for analysis.

We used these data to quantify concentrations of PM_2.5_ outdoors, in the main library, and in the media room where the HEPA cleaners were located. We analyzed separately the periods with and without HEPA cleaners operating. We used a previously published and validated approach to estimate the PM_2.5_
*F_inf_* into the two indoor spaces [9,19,20]. This approach is based on a recursive form of the mass balance model applied to paired hourly indoor and outdoor concentration measurements, see Equation (1). The model uses censoring algorithms to identify and remove the influence of indoor particle sources. 

Equation (1):(1)$$Finf=\frac{Pa}{a+k}$$
where, *F_inf_* is a function of particle penetration efficiency (*P*), the particle removal rate due to diffusion or sedimentation (*k*), and the air exchange rate (*a*) [9].

This study was approved by the Human Research Ethics Committee at the University of Tasmania (H0015006).

## 3. Results

Over the five-month study, there was a wide range of hourly outdoor PM_2.5_ concentrations (median = 31 µg/m^3^, 5th–95th percentile = 2–113 µg/m^3^). Outdoor air quality was poorest during bushfires that occurred in November. The 24-h average outdoor PM_2.5_ concentration peaked at nearly 600 µg/m^3^ on November 15th. Across the whole study period, there were 34 days when 24-h average outdoor air PM_2.5_ concentrations exceeded the national standard of 25 µg/m^3^; 20 of those exceedances occurred in November. We observed that the indoor concentrations of PM_2.5_ in the main library during this time of elevated smoke conditions remained substantially lower than outdoor PM_2.5_, with an absolute reduction in 24-average PM_2.5_ of 8–10 µg/m^3^ being achieved (Table 1).

The library HVAC system was installed in 2000 and was most recently serviced in June 2019. It used cardboard throwaway pleated filters that were replaced annually (F5, type 1 class A). The HVAC system operated from 8 a.m. to 7 p.m. with a 10% fresh air intake during operation. The library had a vestibule; however, both the vestibule door and the entrance door were automatic sliding doors, so they could be opened at the same time. The prevailing winds varied over the five months of the study.

Outdoor PM_2.5_ concentrations were consistently higher than indoor measurements (Figure 1). 

When the HEPA cleaner was off (including the smokiest period in November 2019), *F_inf_* in the main library and media room were 0.31 and 0.32, respectively, meaning that compared to the PM_2.5_ concentration outdoors, the concentrations of outdoor-generated PM_2.5_ indoors were reduced by nearly 70%. When the portable HEPA filter air cleaners were in use in the media room, *F_inf_* was further reduced to 0.17 (Table 1), and concentrations in the media room were systematically 34% lower than in the main indoor library area (Figure 2).

## 4. Discussion

We found that a large building with a central air conditioning system provided an accessible public space with cleaner indoor air during a prolonged air pollution event associated with landscape fires. Inside the building, the concentration of outdoor-generated PM_2.5_ was, on average, 70% lower than outside, and the addition of portable HEPA cleaners in a smaller media room provided additional reductions in PM_2.5_ compared to the PM_2.5_ concentration in the main library.

Our results are consistent with a wide range of residential indoor air quality studies that have demonstrated the benefits of operating HEPA cleaners in appropriately sized rooms during periods of elevated outdoor smoke emissions. Reductions in residential indoor PM_2.5_ concentrations have ranged from 52 to 67% in US and Canadian communities impacted by woodsmoke [13,14,15,19,20]. None, to our knowledge, have tested the efficacy of portable air cleaners for improving the indoor air quality of public buildings.

While most studies have evaluated improvements in air quality from the use of HEPA filtration, several have also evaluated health endpoints related to exposure to landscape fire smoke. For example, studies in Sweden and Canada have found clinically relevant improvements in microvascular function, an important risk factor for cardiovascular disease, associated with the use of HEPA filtration in homes located in areas affected by wood-heater-related air pollution [21,22]. These suggest that the provision of cleaner indoor air environments has the potential to protect health.

Our study operated over an extended period of ongoing smoke resulting from a peat fire and bushfires. This allowed us to evaluate a range of elevated outdoor smoke conditions and to assess the efficacy of installing the HEPA cleaners within a smaller room inside the library area. The HEPA cleaners were only deployed for two months due to limited resources. Thus, the air cleaners were not in place during the most extreme smoke concentrations, which occurred in November 2019. 

We unexpectedly found that the infiltration efficiency was higher during periods with less outdoor smoke. It could be that doors were sealed more effectively during the more severe pollution periods, as these periods were also associated with widespread community messaging to reduce exposures. It may also be possible that fewer people ventured outside to visit the library during more intense smoke episodes resulting in less frequent door opening at the library.

Some limitations should be acknowledged. We did not have information on the use of the library doors—how often they were opened and closed, and where they were situated with respect to the prevailing winds. While factors such as the HVAC operation remained consistent throughout the study, meteorological conditions which have been identified in other studies to influence *F_inf_* would have changed between winter and summer. The influence of these factors on pollution infiltration could not therefore be evaluated in our study. Further, we did not have information about library staff behaviours, so we did not know if these were different during more smoky periods compared with less smoky periods or how this could have influenced indoor air quality. This study also focused only on the PM emissions from the fires and did not account for any gaseous emissions which could also be detrimental to health.

The public health response to episodic air pollution events centres on the provision of public communications advising on actions to reduce exposure to air pollution [23]. It has been suggested that simple measures such as staying indoors and keeping doors and windows closed can reduce exposures to these events for short periods of time but is less effective over several days or weeks of reduced air quality [8,24]. Indeed, improved health outcomes, as distinct from reduced air pollution, have only been demonstrated when indoor air filtration is used in addition to remaining indoors [8,21,23]. With widespread or prolonged air pollution episodes, the creation of cleaner air spaces in indoor environments is potentially a simple and practical means to reduce overall exposure to air pollution, and it offers individuals a location to find some respite from smoke exposures. Because population-level health impacts suggest that there are no lower safe thresholds, all reductions in exposures can be expected to provide health protection, even when outdoor concentrations are not extreme. This is especially true for episodes of pollution that persist for many days or weeks, when options such as remaining indoors or relocating to nonpolluted areas become increasingly impractical [23,24,25]. Guidance to seek shelter in public buildings is available from a range of organisations including US EPA [7], Health Canada [18] and British Columbia Centre for Disease Control [17]. Our results suggest that this option could be a practical and useful addition to the suite of interventions and advice currently provided in Australia and elsewhere. Further evaluation of the potential benefits and feasibility of this approach is warranted.

## 5. Conclusions

A carefully selected public building may provide a cleaner indoor air shelter during episodes of elevated fire smoke pollution. Further improvements in indoor air quality within a building can be achieved by selecting appropriately sized HEPA cleaners to operate within the building space, or in a smaller room. Provision of such shelters by jurisdictions requires planning and careful evaluation of potential sites to ensure that the most useful sites can be identified in advance and made available in a timely way when needed.

## Figures and Tables

**Figure 1 ijerph-18-04085-f001:**
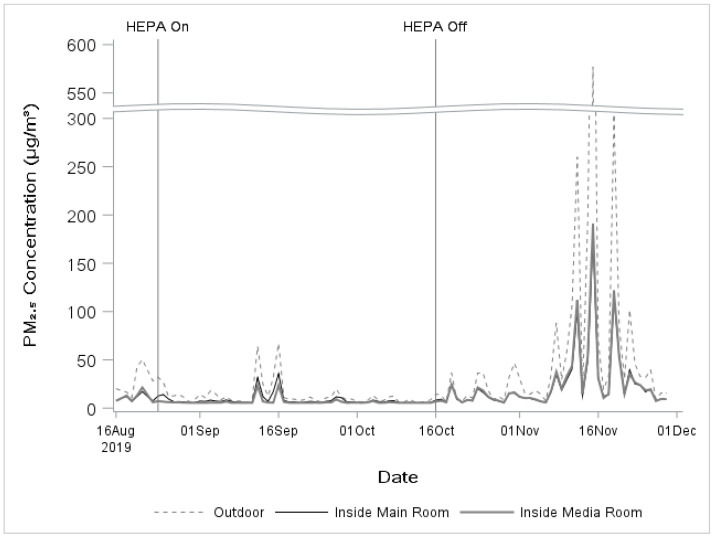
Daily (24-h) average PM_2.5_ over the study period, from the outdoor library, indoor library area and media room SMOG monitors. Note: This figure includes all measurements, but the analysis included only 24-h periods with average outdoor PM_2.5_ concentration > 20 µg/m^3^.

**Figure 2 ijerph-18-04085-f002:**
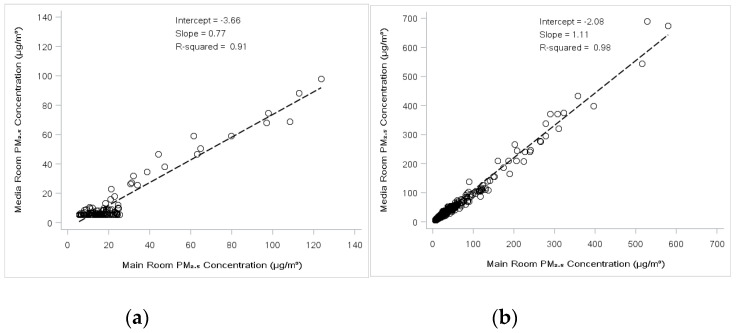
Comparison between indoor library concentrations and media room concentrations with (**a**) and without (**b**) the HEPA cleaner operating.

**Table 1 ijerph-18-04085-t001:** Hourly average indoor and outdoor PM_2.5_ concentrations and infiltration efficiencies (*F_inf_*) calculated with portable HEPA filter air cleaners operating and not operating in the library’s media room.

Date Range (s)	Median Outdoor Conc., µg/m^3^ (25%–75%) ^1^	Indoor Location	Median Indoor Conc., µg/m^3^ (25%–75%) ^1^	Infiltration Efficiency	N ^1^
8/24/19 to 10/16/19	23.3 (12.0–49.1)	Inside library	15.0 (9.8–21.2)	0.45	140
		Media room (HEPA operating)	5.7 (5.5–8.5)	0.17	
8/20/19 to 8/22/19 & 10/18/19 to 11/26/19	30.7 (12.2–85.9)	Inside library	19.6 (9.8–36.2)	0.31	609
		Media room (HEPA not operating)	20.0 (10.5–39.0)	0.32	

^1^ Number of hours included. Used only hours with paired indoor-outdoor data.

## Data Availability

Not applicable.

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
