# Peer review of "Can Public Spaces Effectively Be Used as Cleaner Indoor Air Shelters during Extreme Smoke Events?"

_ijerph, 2021, doi:10.3390/ijerph18084085_

Round 1

Reviewer 1 Report

This is an interesting paper which investigated the potential use of public spaces as cleaner indoor air shelters based on the case study carefully conducted in the library in Port Macquarie, MSW, Australia. Although the results shown in this paper are practically informative, they do not advance our scientific knowledge of the subject sufficiently in the present form.

Please improve following points for the merit of publication.

  • Authors should clearly describe the definition of the infiltration efficiency and explain how to obtain the efficiency using outdoor and indoor concentrations measured by the sensors.
  • Before discussing the effect of the HEPA cleaner using the infiltration efficiency, the values should be discussed with other factors such as outdoor concentration level, operation of the HVAC system (8am-7pm), weather condition and so on. Since the data shown in Table 1 are not sufficient to discuss, I feel difficulty to evaluate this work. Please present more detailed results on the infiltration efficiency. 

Author Response

This is an interesting paper which investigated the potential use of public spaces as cleaner indoor air shelters based on the case study carefully conducted in the library in Port Macquarie, MSW, Australia. Although the results shown in this paper are practically informative, they do not advance our scientific knowledge of the subject sufficiently in the present form.

We have yet to identify any peer-reviewed literature where public buildings have been evaluated for their ability to provide cleaner indoor air environments during extreme smoke events.

Please improve following points for the merit of publication.

Authors should clearly describe the definition of the infiltration efficiency and explain how to obtain the efficiency using outdoor and indoor concentrations measured by the sensors.

The following text explaining the infiltration efficiency was included in the introduction – ‘The infiltration efficiency of outdoor air pollution (Finf) is defined as the fraction of the outdoor concentration that penetrates indoors and remains suspended under steady-state conditions. Finf depends on the particle penetration efficiency, the air exchange rate, and the deposition rate.

Before discussing the effect of the HEPA cleaner using the infiltration efficiency, the values should be discussed with other factors such as outdoor concentration level, operation of the HVAC system (8am-7pm), weather condition and so on. Since the data shown in Table 1 are not sufficient to discuss, I feel difficulty to evaluate this work. Please present more detailed results on the infiltration efficiency.

We have incorporated these concerns into the discussion and limitations – ‘While factors such as the HVAC operation remained consistent throughout the study, meteorological conditions which have been identified in other studies to influence Finf would have changed between winter and summer. The influence of these factors on pollution infiltration could not therefore be evaluated in our study.’

Reviewer 2 Report

Review of ijerph-1169670-peer-review-v1. Can public spaces effectively be used as cleaner indoor air shelters…

Abstract

The abstract might be clearer if the term “air conditioning: was better defined. Specifically, did the air conditioners in this study include filtration and if so, what kind?

“HEPA cleaners” should probably be “HEPA air cleaners” to avoid any confusion with HEPA vacuums.

Does “infiltration efficiency” mean that PM2.5 levels were 30% lower than outside or that they were 30% lower pre- and post? Maybe define the term in the abstract?

The keywords probably should probably include HEPA, PM2.5 and “disaster”

Line 51. Although the focus of this article is on outdoor PM2.5, perhaps indoor sources (cooking, smoking) might be mentioned.

Table 1. What does the far-right hand column labelled “N” mean? Number of readings?

It might help to provide some guidance on maintenance of HEPA air cleaners, and also what “appropriately sized” HEPA air cleaners means.

Thank you for a very well done paper.

Author Response

Abstract

The abstract might be clearer if the term “air conditioning: was better defined. Specifically, did the air conditioners in this study include filtration and if so, what kind?

We have added the descriptor ‘central’ to clarify this statement. Details of the central air conditioning filters are included in the results section.

“HEPA cleaners” should probably be “HEPA air cleaners” to avoid any confusion with HEPA vacuums.

We have included the full term for HEPA in the abstract - high efficiency particulate air (HEPA). The inclusion of “air” is redundant as it is part of the full term.

Does “infiltration efficiency” mean that PM2.5 levels were 30% lower than outside or that they were 30% lower pre- and post? Maybe define the term in the abstract?

We have clarified this statement by using the same terminology as found in the results section – ‘meaning that compared to the PM2.5 concentration outdoors the concentrations of out-door-generated PM2.5 indoors were reduced by nearly 70%.’

The keywords probably should probably include HEPA, PM2.5 and “disaster”

Included as recommended.

Line 51. Although the focus of this article is on outdoor PM2.5, perhaps indoor sources (cooking, smoking) might be mentioned.

Due to the nature of the building (a library) we had no indoor sources. In addition, the method for estimating infiltration efficiency involves identifying and censoring time periods when the indoor concentration was influenced by indoor sources. This method has been used previously to estimate infiltration efficiencies in homes, where indoor emissions would generally be greater than those in a library.

Table 1. What does the far-right hand column labelled “N” mean? Number of readings?

The footnote has been changed to explain in more detail

It might help to provide some guidance on maintenance of HEPA air cleaners, and also what “appropriately sized” HEPA air cleaners means.

We have expanded the methods section to explain the HEPA cleaner constraints – ‘The selection of the HEPA cleaners and the space to be cleaned, needs to account for their operational capacity. This therefore, restricted their use to a smaller room within the main library.’

Thank you for a very well done paper.

Reviewer 3 Report

This presented paper examines the protective function of filter, ventilation and air conditioning technology in the event of severe environmental pollution (e.g. fires and other particle emissions). The current state of knowledge and technology is adequately presented. The causes and goals of the investigations are adequately described. The paper is limited to particle emissions, gases and their effects are not taken into account. This could lead to wrong conclusions.

REMARKS:

Line 148/149: The figure caption should be placed under the picture on the next page.

Line 165/166: The figure caption should be placed under the picture on the next page.

Author Response

This presented paper examines the protective function of filter, ventilation and air conditioning technology in the event of severe environmental pollution (e.g. fires and other particle emissions). The current state of knowledge and technology is adequately presented. The causes and goals of the investigations are adequately described. The paper is limited to particle emissions, gases and their effects are not taken into account. This could lead to wrong conclusions.

The reviewer makes a pertinent point regarding the lack of data related to gaseous exposures. We have added this as a limitation – ‘This study also focused only on the PM emissions from the fires and did not account for any gaseous emissions which could be detrimental to health.’

REMARKS:

Line 148/149: The figure caption should be placed under the picture on the next page.

Line 165/166: The figure caption should be placed under the picture on the next page.

Changed as recommended.

Round 2

Reviewer 1 Report

I had read previous papers by Allen et al. [references 9 and 20, previous 7 and 18] and found originality and novelty for addressing the infiltration factor. So, I asked authors to add more descriptions about it to emphasis the value of this work. For the convenience of readers, please consider the additional explanation about Fint by mathematical expression in the section of Materials and Methods.

Author Response

As requested, the equation has been included in the methods section - 

This approach is based on a recursive form of the mass balance model applied to paired hourly indoor and outdoor concentration measurements, see Equation 1. The model uses censoring algorithms to identify and remove the influence of indoor particle sources.

Equation 1: Finf = Pa / a +k

Where, Finf is a function of particle penetration efficiency (P), the particle removal rate due to diffusion or sedimentation (k), and the air exchange rate (a)[9].